# Cost analysis of an intrapartum quality improvement package for improving preterm survival and reinforcing best practices in Kenya and Uganda

**Carolyn Smith Hughes**[1]*, **Elizabeth Butrick**[1], **Juliana Namutundu**[2], **Easter Olwanda**[3], **Phelgona Otieno**[3], **Peter Waiswa**[2], **Dilys Walker**[1], **James G. Kahn**[1]

**1** Institute for Global Health Sciences, University of California San Francisco, San Francisco, California, United States of America, **2** School of Public Health, Makerere University, Kampala, Uganda, **3** Kenya Medical Research Institute, Nairobi, Kenya

* carolyn.smithhughes@ucsf.edu

**Data Availability Statement:** PTBi cost analysis data files are available from the Dryad Digital

## Abstract

### Introduction

Preterm birth is a leading cause of under-5 mortality, with the greatest burden in lower-resource settings. Strategies to improve preterm survival have been tested, but strategy costs are less understood. We estimate costs of a highly effective Preterm Birth Initiative (PTBi) intrapartum intervention package (data strengthening, WHO Safe Childbirth Checklist, simulation and team training, quality improvement collaboratives) and active control (data strengthening, Safe Childbirth Checklist).

### Methods

In our analysis, we estimated costs incremental to current cost of intrapartum care (in 2020 $US) for the PTBi intervention package and active control in Kenya and Uganda. We costed the intervention package and control in two scenarios: 1) non-research implementation costs as observed in the PTBi study (Scenario 1, mix of public and private inputs), and 2) hypothetical costs for a model of implementation into Ministry of Health programming (Scenario 2, mostly public inputs). Using a healthcare system perspective, we employed micro-costing of personnel, supplies, physical space, and travel, including 3 sequential phases: program planning/adaptation (9 months); high-intensity implementation (15 months); lower-intensity maintenance (annual). One-way sensitivity analyses explored the effects of uncertainty in Scenario 2.

### Results

Scenario 1 PTBi package total costs were $1.11M in Kenya ($48.13/birth) and $0.74M in Uganda ($17.19/birth). Scenario 2 total costs were $0.86M in Kenya ($23.91/birth) and $0.28M in Uganda ($5.47/birth); annual maintenance phase costs per birth were $16.36 in Kenya and $3.47 in Uganda. In each scenario and country, personnel made up at least 72%

Repository at doi:10.7272/Q64X562W (https://doi.org/10.7272/Q64X562W).

**Funding:** The PTBi study, including the cost analysis, was funded by the Bill & Melinda Gates Foundation. The grant was received by Dr. Dilys Walker (DW, grant number OPP1107312; https://www.gatesfoundation.org/). The funder had no role in study design, data collection and analysis, decision to publish, or preparation of the manuscript.

**Competing interests:** DW is cofounder of PRONTO International. This does not alter our adherence to PLOS ONE policies on sharing data and materials. The other authors declare no competing interests.

of total PTBi package costs. Total Scenario 2 costs in Uganda were consistently one-third those of Kenya, largely driven by differences in facility delivery volume and personnel salaries.

## Conclusions

If taken up and implemented, the PTBi package has the potential to save preterm lives, with potential steady-state (maintenance) costs that would be roughly 5–15% of total per-birth healthcare costs in Uganda and Kenya.

## Introduction

Each year, about 15 million babies are born prematurely around the world [1], and 1 million infants and children will die due to prematurity-related complications [2]. In addition, more than 2 million stillbirths occur globally per year [3]. While stillbirth rates are generally declining, preterm birth rates are increasing, and the burden of preterm birth is highest in lower-resource settings, including sub-Saharan Africa [1, 3, 4]. Preterm birth and stillbirth lead to significant loss of life, lost productivity, and anguish among surviving family members; these conditions are also associated with substantial disability and healthcare costs [5–7]. Healthcare costs are generally inversely proportional to gestational age, not only during the initial days and weeks after birth, but also later in life [6].

Interventions to prevent stillbirths and improve preterm survival have been developed and tested in high- and low-resource settings alike [6, 8, 9], with lower-tech strategies demonstrating great promise and feasibility in low- and middle-income countries [9]. Given that the largest contribution to both stillbirths and early neonatal deaths occurs around the time of birth, this focus on intrapartum care is critical [10]. One such strategy to prevent stillbirth and improve preterm survival is the Preterm Birth Initiative East Africa (PTBi) package of interventions, which focuses on reinforcement of evidence-based practices to improve the quality of intrapartum and immediate newborn care provided to mothers and their infants [11].

The PTBi package of interventions (and other strategies to prevent preterm birth and promote neonatal survival) has been evaluated for effectiveness; however, data on the cost of quality improvement programs are limited [12, 13]. Understanding the start-up and longer-term costs and resources associated with integration of these interventions into current maternity and obstetric care practices may help inform resource allocation for these life-saving strategies.

Data regarding costs of maternity care are limited and estimates vary substantially by country, health facility type/delivery volume, and source/reference [14]. For example, one analysis estimated that health system costs per delivery/birth in sub-Saharan African countries range from $8-$73 USD for a vaginal delivery, $80-$562 per cesarean delivery [15]. However, a 2015 analysis found an average cost per birth of $106 USD in Kenya and $75 USD in Uganda [16]. The estimated cost of care for low-birthweight infants is estimated to be $514 per infant [15]. Understanding costs associated with the PTBi intervention in Kenya and Uganda may add to the knowledge base and support future programmatic and resource planning.

The goals of our study were 1) to calculate the implementation-level (ie, non-research) costs associated with the PTBi program in Kenya and Uganda, as observed in the PTBi study, and 2) to estimate costs associated with public-sector planning, implementation, and maintenance of the package in a non-study, hypothetical, Ministry of Health (MOH) model. Modeled costs in a non-study/MOH setting may be more relevant to decision-makers to inform

resource allocation; as such, our analysis focuses primarily on estimates for the cost of public sector implementation. To our knowledge, this is among the first studies to assess the costs of a complex, multicomponent intrapartum quality improvement package in sub-Saharan Africa.

## Methods

### Overview

Through the PTBi study in Kenya and Uganda, we estimated the per-birth (term and preterm; live born and stillborn; in $2020 US) costs associated with the PTBi package of interventions compared to a more minimal active control. Using a healthcare system perspective, we analyzed incremental costs–costs that could be added to the current cost of obstetric care–associated with the PTBi package and the active control using an activity-based micro-costing approach. Our analysis includes incremental costs for all deliveries with an estimated gestational age of 28 weeks or higher, regardless of delivery outcome (ie, both live births and stillbirths). We calculated costs for both the package of interventions and active control through two scenarios: Implementation as observed in the PTBi study (excluding purely research costs) and a hypothetical model led by the Kenyan and Ugandan Ministries of Health.

### Study setting

The PTBi study was a cluster-randomized controlled trial implemented in 17 health facilities in Migori County, Kenya and 6 health facilities in Busoga Region, Uganda [17] from 2016 to 2019. The study was approved by the Kenya Medical Research Institute, Makerere University School of Public Health, and the University of California, San Francisco Institutional Review Boards. Detailed descriptions of the study have been published previously [17, 18]. This cost analysis did not involve direct or indirect interaction with the patients (mothers or infants) or review of patient records/information from the effectiveness portion of the main PTBi study.

In the PTBi study, the comprehensive PTBi package of interventions was demonstrated to reduce the odds of stillbirth and early neonatal death among premature infants by 34% by improving intrapartum and immediate newborn care [11]. Thorough reviews of the implementation of the PTBi package found that the full package and its components were feasible, replicable, flexible in the face of challenges, and fostered relationships within and among health facilities [18, 19]. It was also found to have strong potential for "local ownership", due in part to the process of intervention selection and implementation [18], as detailed in the following section.

### PTBi intervention package and control

The PTBi package of interventions included 4 components: data strengthening, a modified WHO Safe Childbirth Checklist (mSCC), PRONTO simulation and team training [20, 21], and Quality Improvement (QI) Collaboratives [11, 17]. In the PTBi study, selection of the interventions was based on published literature reviews, robust assessments of health facilities in Kenya and Uganda, and several meetings with local stakeholders [11, 17, 18, 22]. Through this process, PTBi study investigators found that while global and local national guidelines were in place in Kenya and Uganda, actual care delivered in facilities was not consistently per these guidelines. Further, measuring and tracking of health data and quality indicators were deficient, and health workers often lacked the clinical skills and training to identify and manage obstetric emergencies and preterm birth. Based on this review, the 4 PTBi package components were identified as mutually-reinforcing quality improvement interventions that were aligned with WHO and local guidelines and could be tailored to address quality of clinical

obstetric and neonatal care, with a focus on preterm birth [11, 17, 18]. While each of the intervention components had previously been evaluated individually for their effectiveness in other settings, some findings have been mixed, the 4 components had not been bundled, and none of the interventions were in place in Kenya or Uganda [17, 18].

From a clinical care perspective, this PTBi package was designed to reinforce evidence-based practices for maternal and newborn care around the time of birth, including provision of antenatal corticosteroids; immediate skin-to-skin contact and kangaroo mother care (KMC); newborn resuscitation; and appropriate feeding [11, 18]. It was also designed to strengthen provider skills in maternity and neonatal care, and to promote teamwork and communication among health workers [17]. Beyond direct clinical care, the package also aimed to promote data use for decision making–including gestational age estimation and tracking of key indicators–and other quality improvement strategies.

The active control was also selected through the above-mentioned process. Data strengthening and the WHO mSCC have been outlined by WHO and other local and national organizations as important interventions to better track and compare health outcomes, help improve care during childbirth, and promote the use of evidence-based practices [1, 9, 23, 24]. Intervention components are described further in Table 1, and detailed descriptions of the intervention components are available in study publications [11, 17, 18].

In the PTBi study, this package of four interventions was compared to a more minimal active control that included data strengthening and mSCC [17]. Activities and resources for data strengthening and mSCC are the same between the package and control; however, because all four components of the PTBi package are complementary and mutually reinforcing, there is efficiency in delivering all 4 interventions in the package versus 2 in the control. For example, transportation related to monitoring of activities for all 4 interventions would be similar to/the same as that for data strengthening and mSCC alone. We chose to cost the active control for the following reasons: the PTBi package and active control were compared in the effectiveness study; the package and control were each implemented into, and as a supplement to, the existing current practice; and because the active control components are aligned to WHO and local recommendations.

As described above, the selection, design, and implementation of the PTBi package of interventions and control was a collaborative effort by researchers and clinicians in Kenya, Uganda, and the US, and was based on specific in-country needs [18]. Individual components of the package– DS, a mSCC, simulation and team training through PRONTO International, and QICs– were tailored to stakeholder-identified priorities and key clinical issues that impact preterm survival in each country [18]. Implementation in other regions, facilities, or settings and for other obstetric and neonatal needs may differ from the PTBi study.

## Modeling implementation scenarios and framework

In our analysis, we costed implementation of the PTBi package and control under two scenarios: 1) implementation-level costs (non-research) as observed in the PTBi study (Scenario 1) and 2) costs for a hypothetical model of implementation by local Ministries of Health (Scenario 2). In Scenario 1, we took care to identify and exclude activities that were exclusively associated with research activities, such as data collection that would only be undertaken in a study setting. However, even after excluding activities and costs associated with research, clinical studies typically reflect prices and intensity levels that are not representative of costs associated with program implementation in the public sector. Because of this, we developed our second scenario to model hypothetical implementation ("ownership") and management by local MOHs in a non-research setting, using mostly public-sector inputs.

**Table 1. Description of PTBi package and control components.**

| Intervention component | Objective | Activities |
|---|---|---|
| Data Strengthening (intervention and control facilities) | Improve gestational age estimation, measurement, and data use for preterm birth and other neonatal and maternal indicators | • Select and review indicators<br>• Use pregnancy wheels and measuring tapes<br>• Create/adapt data dashboards<br>• Perform data quality assessments (DQAs)<br>• Conduct workshops and provide data mentoring<br>• Collect and use data for data indicator monitoring |
| Modified WHO Safe Childbirth Checklist (mSCC) (intervention and control facilities) | Reinforce evidence-based clinical practices and country-specific guidelines during labor and delivery | • Adapt checklist align to local guidelines<br>• Review with experts and stakeholders<br>• Pilot and revise checklist<br>• Use mSCC for all in-facility births, including gestational age assessment |
| PRONTO Simulation and Team Training (intervention facilities only) | Highly realistic hands-on training and mentorship to increase uptake of evidence-based practices, improve clinical care, and bolster team communication | • Adapt clinical curricula focused on obstetric complications, preterm birth, neonatal care<br>• Recruit and train PRONTO Nurse Mentors<br>• Conduct in-facility teamwork exercises and simulations to promote use of standard clinical practices |
| Quality Improvement (QI) Collaboratives (intervention facilities only) | Enable facility staff to track and use indicators to improve care for preterm infants | • Establish facility- and country-based QI teams matched with coaches<br>• QI teams partner to discuss and implement QI projects<br>• Track QI indicators |

DQAs: Data quality assessments; mSCC: modified WHO Safe Childbirth Checklist; QI: Quality improvement.

Table 1 provides a description of the PTBi package of intervention components, key objectives of each component, and activities within each component, as implemented in the PTBi Study (Scenario 1), and as estimated in our hypothetical MOH model (Scenario 2).

In both Scenarios, we calculated costs incremental to the cost of providing intrapartum maternal and newborn care; as such, our findings represent the additional costs necessary to implement the PTBi package or control within the current level of care provided in our analysis setting. Further, costs in this analysis focus on those related to the direct planning, implementation, and maintenance of the PTBi package of interventions; costs do not include indirect or downstream changes in medical care. For example, we included costs for resources (paper, printing) and health worker time completing the mSCC for each delivery, but we did not include the costs associated with changes in the usage of medications or devices following PTBi package implementation.

To understand the costs for both scenarios, we created an "implementation framework" that documented the phases, activities, and key personnel involved in each PTBi package component for both scenarios. Within this framework, we also outlined activities that are shared between intervention components or are not easily distinguishable (eg, initial program planning and stakeholder meetings, and training sessions that encompass several interventions). Descriptions of both scenarios are outlined in Table 2.

In our framework, we outlined three critical and distinct implementation phases associated with both scenarios: 1) program design and planning (6–9 months), 2) high-intensity initial implementation (12–18 months) and 3) annual maintenance. Phases and timing of activities are depicted in Fig 1 and are based on discussions with PTBi study team members. More details on timing of activities for Scenario 2 is available in S1 Fig.

**Table 2. Description of costing scenarios and major inputs.**

| Input category and definition | Scenario 1 *Implementation in PTBi Study* | Scenario 2 *Hypothetical MOH Model* |
|---|---|---|
| **Personnel**<br>Time spent by program staff, specialist consultants, and clinicians to plan, review, implement, and monitor use of interventions in facilities | • Private sector salaries for most/all program personnel<br>• Heavy use of consultants for program development | • Public sector salaries for most program and facility-based personnel<br>• Minimal private sector/consultant personnel |
| **Consumable goods**<br>Single-use clinic and office goods, including supplies used for training<br>**Capital goods**<br>Durable clinic and office goods, including laptops for program specialists | • Mix of study and MOH supply chains | • MOH and lowest public sector prices |
| **Travel**<br>Transportation and lodging for major meetings, trainings, program activities and monitoring | • International, national, and local travel for non-research purposes | • Mostly local and regional travel<br>• Minimal international travel |
| **Program office/conference space**<br>Office space for program-specific activities; costs for offsite meetings, including rented conference rooms. | • Percentage of study office rent<br>• Hotel conference rooms for meetings and training | • Value of office space for program specialists<br>• Limited use of conference rooms, prorated based on percentage of time spent on PTBi package |

MOH: Ministry of Health; PTBi: Preterm Birth Initiative.

Table 2 describes major cost categories, the definitions of these categories, and descriptions of inputs within these categories, specific to Scenarios 1 and 2 within the PTBi cost analysis.

To create this "implementation framework", we used study records, including budget and expenditure documents and training manuals; QI guidelines and WHO recommendations; MOH staffing and hiring documents and organograms; and facility assessment records. Our framework: 1) outlines key activities associated with each intervention, 2) maps the timing and steps associated with these activities, 3) maps the personnel and resources critical to the successful application of these activities, and 4) documents how activities vary in intensity over time. After initial document review, we conducted interviews with key study personnel, including program managers, consultants, costing experts, and clinicians to further refine our framework.

## Activity-based micro-costing

For both scenarios, we used empirical bottom-up activity-based micro-costing to estimate incremental costs associated with activities within each phase of the PTBi package and control. Key activities included stakeholder meetings to discuss and adapt intervention components; component-specific training (initial training and refresher training) for program-based and facility-based staff and clinicians; program management by MOH MNCH staff; and time spent by program and facility staff "using" the interventions (eg, participating in PRONTO simulations in facilities, attending QIC meetings, and filling out the mSCC for each birth). Activities are further described in Table 1 and Figs 1 and S1.

Our main cost input categories included costs associated with personnel, consumable goods, capital equipment, travel, and program office/conference space. Input categories are outlined in Table 2 and are described in detail in the following sections. Scenario 2 select unit prices for each major input category are in Table 3; unit prices for Scenario 1 are available upon reasonable request.

We tabulated total costs per country, phase, intervention component, health/birth facility, and birth (regardless of gestational age, including live births and stillbirths) and report in $US 2020 (cost data were adjusted for inflation where appropriate for all inputs). When prices and costs were derived from PTBi study invoices and ledgers (eg, invoices for clinic supplies used for PRONTO simulations or office supplies for program implementation), in-country prices and costs were converted from Kenyan Shillings and Ugandan dollars into $US using the exchange rate for the month of the invoice/ledger. Where prices and costs were derived from annual statements or estimates (eg, MOH personnel salaries), costs were converted using the

| Phase 1 | Phase 2 | Phase 3 |
|---|---|---|
| **Planning**<br>**6-9 months** | **Startup**<br>**12-18 months** | **Maintenance**<br>**Annual** |
| *All steps and activities prior to implementation in facilities* | *Initial intensive implementation in facilities* | *Ongoing DS, mSCC, PRONTO, and QI activities* |
| **Initial planning**<br>• High-level planning meetings<br>• Indicator selection<br>• Curriculum development/adaptation*<br>• Adaptation of mSCC and review<br><br>**Preparation for deployment**<br>• Hiring personnel<br>• Facility selection, assessments<br>• Piloting tools<br>• TOT for local mentors, other trainers*<br>• Preparation for facility-level training<br>• Purchasing goods needed for planning and initial startup | **Training**<br>• Training of facility staff and clinicians for DS, mSCC, and QI activities<br>• PRONTO introductory training<br><br>**Use of interventions**<br>• PRONTO mentorship with simulation and team training<br>• Initiation of DS, mSCC, and QI activities<br>• DS data collection, reporting, and related QI activities<br>• Data quality assessments (more frequent)<br>• QI collaboratives<br>• Use of mSCC<br>• Purchasing goods as needed | **Intervention activities**<br>• Ongoing QI activities, including indicator tracking<br>• Team review of facility data and indicators (QI)<br>• Data quality assessments (less frequent)<br>• QI collaboratives<br>• Use of mSCC<br>• Ongoing PRONTO simulation and teamwork activities<br>• Purchasing/replacement of goods as needed<br>• Annual refresher training for all personnel*, and refresher TOT for nurse mentors |

**Fig 1. Implementation (non-research) phases and activities (Scenarios 1 and 2).** DS: data strengthening; mSCC: modified WHO safe childbirth checklist; QI: Quality improvement; TOT: training of trainers. *Includes activities related to DS, mSCC, PRONTO, and QI. Fig 1 describes the phases of implementation for the PTBi package of interventions. Each phase includes timing and critical activities to be undertaken throughout the phase. This applies to both Scenarios 1 and 2, though annual maintenance costs were not observed or included in Scenario 1.

average exchange rate for the year of the estimate. All prices and costs were then converted to 2020 $US to account for inflation using annual within-country inflation rates.

Costs for each phase were tabulated for all activities within each phase; costs per facility were tabulated by dividing total country costs by the number of facilities in each country. Costs per birth were tabulated by dividing total country costs by the total number of births that occurred in facilities over the duration of implementation (monthly facility delivery volumes for all facilities, multiplied by the 27 of months of implementation [average 15 months of initial high-intensity implementation and 12 months maintenance]).

**Table 3. Select input prices and units for PTBi package of interventions (Scenario 2).**

| | Scenario 2 (Hypothetical MOH Model) | | | |
| --- | --- | --- | --- | --- |
| | Kenya | | Uganda | |
| **Personnel** | Unit price | Units | Unit price | Units |
| **Program-based** | (per month, incl benefits) | (months) | (per month, incl benefits) | (months) |
| County or regional/district MNCH coordinator | $2,181 | 3.5 | $807 | 3.5 |
| Sub-county or associate MNCH coordinator | $1,211 | 10 | $718 | 10 |
| Senior obstetrician | $3,656 | 4.5 | $576 | 4 |
| Senior pediatrician | $3,656 | 4.5 | $576 | 4 |
| Statistician/biostatistician | $1,203 | 7 | $658 | 6.5 |
| Data quality/mSCC specialist | $870 | 42 | $386 | 25 |
| DHIS/HMIS specialist | $1,203 | 5.5 | $658 | 5.5 |
| PRONTO/QI specialist | $1,286 | 65 | $386 | 31 |
| **Facility-based** | | | | |
| Principal medical officer | $2,505 | 30 | $389 | 23 |
| Maternity/newborn unit in-charge | $1,534 | 48 | $389 | 33 |
| PRONTO Nurse Mentor | $1,211 | 85 | $386 | 35 |
| Registered nurse or midwife | $840 | 204 | $210 | 158 |
| Health records clerk | $472 | 72 | $121 | 34 |
| **Consumable goods** | Unit price | Units | Unit price | Units |
| **Office supplies and paper-based goods** | | | | |
| PRONTO Manuals–Professional printing (per unit) | $38 | 20 | $38 | 10 |
| PRONTO Manuals–Local printing (per unit) | $6 | 341 | $5 | 212 |
| DS, mSCC, and QI training packets (per unit) | $2 | 403 | $2 | 197 |
| mSCC (paper and printing, 100 per unit) | $18 | 372 | $15 | 507 |
| **Clinic goods** | | | | |
| Single-use PRONTO supplies (per mentor per year) | $214 | 40 | $198 | 18 |
| **Capital goods (per unit)** | Unit price | Units | Unit price | Units |
| Computer | $998 | 4 | $821 | 2 |
| Office desk and chair | $115 | 4 | $91 | 2 |
| Projector | $232 | 1 | $137 | 1 |
| Neonatal resuscitation mannequin | $85 | 32 | $85 | 18 |
| Bag-valve masks (neonatal/preterm) | $46 | 20 | $26 | 7 |
| **Program office/conference space** | Unit price | Units | Unit price | Units |
| Hotel conference room (per day) | $628 | 6 | $343 | 6 |
| Food/beverage—offsite meetings (per person/day) | $18 | 128 | $15 | 116 |
| Value of office space for specialists (per month) | $105 | 132 | $28 | 72 |
| **Travel** | Unit price | Units | Unit price | Units |
| Airfare: US to in-country (return trip) | $2,000 | 3 | $1,850 | 3 |
| Airfare: Within East Africa (return trip) | $215 | 21 | $310 | 18 |
| Hotel accommodation and food (per night) | $65 | 102 | $55 | 51 |
| Petrol/vehicle expenses—for local travel (return trip) | $9 | 1280 | $11 | 537 |

DS: data strengthening; MNCH: Maternal, newborn, and child health; mSCC: modified WHO safe childbirth checklist; PTBi: Preterm Birth Initiative; QI: Quality improvement.

Table 3 provides unit prices and units associated with select items/resources within main cost input categories (personnel, consumable goods, capital goods, facilities, and travel) for Scenario 2. Due to confidentiality, Scenario 1 unit prices have been withheld from publication. Data on personnel salaries were collected from publicly available records regarding health sector pay and were matched to appropriate cadres through in-country data validation; prices for consumable goods were based on in-country MOH price lists for office and clinic goods; prices for capital goods were gathered from publicly available price lists and based on local fair market value; conference space and food/beverage were based on data from the PTBi Study; travel prices were based on current market value at the time of data collection and validated versus previous records (late 2020).

**Key personnel.** We estimated Scenario 1 program-based personnel costs as observed in the PTBi study. Data sources included salary records, budget and expenditure records, timelines, and other project records for the PTBi study; such costs are representative of the private sector. For Scenario 1 facility-based personnel and all non-consultant Scenario 2 personnel, salary data were extracted from published pay scales for program-based, specialist, and facility-based workers that would be involved in the planning and implementation of the interventions in the public sector [25, 26]. We linked the pay scales for each worker cadre to job descriptions, required credentials, and roles and responsibilities of those cadres in recruitment, hiring, and employment guidelines [27–30]. Select personnel salaries and units are in Table 3.

We quantified personnel effort for each facility-based cadre and program-based staff in both scenarios using study records and interviews with program experts. We estimated the amount of time spent on each intervention component and activity, taking care to exclude activities specifically associated with clinical research. Personnel time included time spent by study coordinators and program managers to lead, implement, and monitor intervention activities; clinical specialists and supervisors providing oversight and expertise on specific topics; PRONTO training specialists and Quality Improvement experts (specialist consultants); data specialists and health records clerks receiving training and tracking indicators; and health workers receiving training in facilities and participating in intervention activities. We estimated that, on average, between 8 and 15 health workers would receive at least some level of training or exposure to the PTBi intervention and control per facility [22].

In the PTBi study, new clinical providers were not added to the health facilities within the study; as such, we assumed that new clinical providers would not be hired specifically for the PTBi intervention or control in our cost analysis, and thus did not include costs associated with new clinical providers in either Scenario 1 or 2. We did, however, account for the need for additional training regarding PTBi intervention and control components for health workers who are hired to replace other staff as part of the natural hiring process (ie, hiring staff to replace others due to turnover, promotion, retirement, or other circumstances).

In Scenario 1, PTBi study staff undertook all program management activities; via interviews, we estimated the percentage of personnel time spent on general program activities and for each PTBi package component. In Scenario 2, we assumed that program management would be undertaken by local MOH personnel working on existing MNCH platforms/programs. However, we also estimated that two new roles would need to be created to implement PTBi activities: 1) a data specialist role that would be responsible for overseeing DS and mSCC activities, and a 2) QI specialist role that would be responsible for overseeing PRONTO and QI activities. We used MOH organograms to understand reporting, hierarchies, and appropriate pay scales, and supervision for these roles. We included onboarding and supervisory time in our personnel cost estimates for these roles, and we assumed that one specialist within each role would be needed for every 5–8 facilities, depending on facility delivery volume (eg, one person for 4–6 large facilities or 8–9 small facilities).

PRONTO Nurse Mentors are experts in maternity and neonatal care and are essential to the successful implementation of PRONTO activities [11, 18, 21, 31, 32]. In the PTBi study (Scenario 1), several mentors were hired in each country to undertake all PRONTO activities, including traveling to each study facility to lead simulations and team training activities, and provide bedside clinical mentoring [11, 17, 18]. In Scenario 2, we modeled the training of one nurse, midwife, or medical officer within each health facility to undertake facility-based PRONTO activities; this model was developed based on discussions with in-country experts with knowledge of sustainable practices in MOH training activities. Costs associated with training nurse mentors (Training of Trainers) was included in both scenarios.

In the PTBi study, the PRONTO and QI activities required the intensive involvement of private-sector experts and specialists for the design and implementation of simulation/team training and QI activities [18]. Scenario 1 costs reflect this involvement. Scenario 2 assumes minimal adaptation (not full development) of these existing interventions. Therefore, Scenario 2 costs reflect less intensive consultation with experts, their involvement in stakeholder meetings, their time spent adapting existing interventions (assuming minimal adjustments), and their role in training of trainers.

**Consumable and capital goods.** Consumable goods in our scenarios included office and simulation supplies used for the package of interventions and control. Office supplies included printed training materials, checklists (mSCC), and goods used for QI activities (eg, paper, markers, and materials for QICs). Simulation supplies included syringes, gauze, dipsticks, and other clinical goods used for PRONTO training activities. Unit prices and quantities for all consumable goods for Scenario 1 were extracted from PTBi study budget and expenditure records; Scenario 2 unit prices were extracted from published price lists and order forms [33, 34], and quantities were based on those from the PTBi study. Select unit prices are in Table 3.

We identified the capital goods (including quantities and replacement frequencies) required to implement and maintain interventions through a review of study records and interviews with study staff. Capital goods in both scenarios included laptops and office furniture for program coordinators/managers, projectors for training and ongoing QICs, and the PRONTOPack[TM] simulation training kit and neonatal resuscitation mannequins for PRONTO activities. We assumed replacement laptops, projectors, and PRONTOPacks[TM] every 3–5 years. Because Scenario 2 assumes integration of the PTBi package into existing MNCH programming, we assumed that laptops and office furniture would only be purchased for the new roles described in the personnel section. The value of capital office goods for existing MNCH program staff was prorated based on the percent effort used to incorporate the package of interventions into the local MNCH platform. We also assumed that all PRONTOPack[TM] components in Scenario 2 would be purchased in-country. Unit prices and quantities in Scenario 1 were extracted from study budget and expenditure files; unit prices for Scenario 2 capital goods were extracted from MOH price lists for most items, with PTBi study prices for specialized goods that were not available in MOH price lists.

While the PTBi package of interventions does not include the use of clinical supplies or capital goods that are not already part of essential maternity and newborn care in-country, facility assessments in the PTBi study found that some facilities did not have all the necessary goods required to provide care per guidelines [22]. As such, for both scenarios, we included costs for some basic clinical goods, such as neonatal bag valve masks and tubing, prorated based on the percentage of PTBi study facilities that were lacking these goods at baseline.

**Travel and other costs.** We estimated costs for transportation in each scenario, including vehicle costs for travel to and from facilities for regular intervention activities. We also estimated consultant travel costs, which included limited regional and international airfare, ground transportation, and lodging for multi-day activities, such as training of PRONTO mentors. Travel costs were calculated in Scenario 1 were using on observed prices and quantities in study documents and invoices. Scenario 2 costs were calculated using local price estimates collected in-country by local health economists in March 2020; travel units were based on our analysis of program activities that would require travel for consultants, program staff, and limited travel for select in-facility workers.

We estimated costs associated with program office space based on study office rent in the PTBi study (Scenario 1; prorated for the average amount of time personnel within those offices spent working on implementation activities) and hotel conference rooms rented for major meetings and training sessions, including food and beverages provided during offsite

meetings. In Scenario 2, we estimated the value of program office space that would be used within existing MOH program offices for program-based staff and limited use of conference rooms for annual, offsite MNCH planning meetings, prorated for the percentage of the meeting that would be used to discuss PTBi activities. Scenario 2 assumes that all training activities, PRONTO sessions, and QICs would take place within existing space in health facilities; therefore we did not include health facility space costs for these activities.

## Outcomes and assumptions

For Scenario 1 (costs observed in the PTBi study), we calculated total incremental costs associated with package implementation in 9 health facilities in Kenya and 4 in Uganda; we also

**Table 4. Scenario 2 (Hypothetical MOH model) assumptions and rationale.**

| | Assumption | Rationale |
|---|---|---|
| 1 | PTBi package supplements existing MNCH services (no displacement), with PTBi activities costed, even for existing staff | Package reinforces existing best practices, does not introduce new clinical interventions, but adds training and monitoring activities |
| 2 | Integration of PTBi package activities within the individual region in each country | Health programming is implemented at county and sub-county levels in Kenya and regional and district levels in Uganda |
| 3a | Political will, understanding of need for interventions, and interest in PTBi package | PTBi package was found to be acceptable among health workers and was demonstrated to significantly improve preterm survival |
| 3b | While outside donors are critical to the funding of public sector care in each country, we did not assume a specific non-MOH funder and did not include costs associated with donor compliance or management of other donor-specific activities | Costs associated with donor compliance, coordination, communication, and other activities are specific to each donor and were therefore not included in Scenario 2 |
| 4 | Adaptation (not full development) of intervention components | Interventions developed through PTBi Study were demonstrated to be highly effective and would likely require minimal adaptation for further implementation. Inclusion of new EmONC content would incur curricular and mentor training costs (not counted in this analysis) |
| 5 | Public-sector funding through local MOH programming | MOHs are the single largest provider of care in each country |
| 6 | Implementation of most activities by MOH staff and clinicians | Reflects current MNCH activities in public sector |
| 7 | Scenario costs are based on public-sector prices and salaries paid by MOHs | Implementation of activities and purchasing of goods would be done through MOHs |
| 8 | Total costs include time spent by facility-based staff in training and while using interventions | Allows for estimation of full value of package of interventions and opportunity costs |
| 9 | No patient- or community-level costs | Interventions are implemented within health/birth facilities and reinforce existing best practices and guidelines |
| 10 | No purchasing or use of clinic goods or training supplies that cannot be procured within each country | The PTBi package does not require the implementation or purchasing of clinical interventions, goods, or tools that are not accessible in-country |
| 11 | No costs associated with creation or rental of new health facility space | Package is implemented in existing health facilities, does not require additional health facility space |

MNCH: Maternal, newborn, and child health; MOH: Ministry of Health; PTBi: Preterm Birth Initiative.
Table 4 outlines key assumptions for Scenario 2 and the rationale for these assumptions. Some assumptions are speculative, however, they were developed based on qualitative discussions with experts in public health programming in each country.

calculated total costs for 8 control facilities in Kenya and 2 control facilities in Uganda. In our hypothetical MOH model (Scenario 2), we calculated total incremental costs associated with the package of interventions and active control in all 17 facilities in Kenya and all 6 facilities in Uganda. For both scenarios, we calculated total program costs, cost per facility, and cost per birth for each the package of interventions and control. We also tabulated the percent contributions of by input category and intervention component for the PTBi package.

Costs per facility were based on the number of facilities in each country. Cost per birth was based on number of facilities, birth volumes per month per facility, and duration of implementation. Kenyan facilities had volumes of 30–360 deliveries per month, with most facilities having fewer than 100 deliveries per month. Ugandan facilities had volumes of 90–585 deliveries per month, with most facilities having more than 150 deliveries per month. Total costs included 27 months of in-facility activities across phases 2 and 3.

For Scenario 2, we relied on several key assumptions, which were identified through interviews with in-country stakeholders. For example, we assumed that all intervention and control activities would be fully integrated within existing MOH MNCH activities. As such, we also assumed that majority of activities for each phase would be managed and implemented by MOH MNCH program leadership and staff within each geographic area. Other key assumptions and rationale are described in Table 4, and more detailed assumptions are in S2 Fig.

## Sensitivity analyses

We conducted sensitivity analyses to understand the impact of uncertainty in Scenario 2, including higher intensity of program planning activities and intervention adaptation; higher intensity of QI and PRONTO activities, including more frequent refresher training; hiring/use of "centralized" nurse mentors instead of facility-based nurse mentors; and differences in facility size and worker composition within facilities.

## Results

### Scenario 1

For Scenario 1, we calculated costs associated with package implementation in 9 health facilities in Kenya and 4 in Uganda, and 8 control facilities in Kenya and 2 control facilities in Uganda, as observed in the PTBi study. Scenario 1 total program costs by country, input type, intervention, facility, and birth are in Table 5. Total non-research implementation costs for the PTBi study was $2.23 million, including intervention and control activities and inputs; PTBi package costs were $1.85 million in total with a cost per birth of $48.13 per birth in Kenya, and cost per birth of $17.19 in Uganda. PTBi study control costs per birth were $15.22 in Kenya and $6.13 in Uganda.

### Scenario 2

In the hypothetical MOH model (Scenario 2), we calculated costs associated with package intervention and control in all 17 facilities in Kenya and all 6 facilities in Uganda. Scenario 2 total program costs by country, input type, intervention, facility, and birth are in Table 5 above. In Scenario 2, total PTBi intervention costs were $0.86 million in Kenya ($50,497 per facility, $23.91 per birth) and $0.28 million in Uganda ($46,871 per facility, $5.47 per birth). Cost per birth in Phase 3 alone–representing steady-state costs after high-intensity implementation–was $16.36 in Kenya and $3.47 in Uganda.

Scenario 2 costs per birth were 50% lower in Kenya and 68% lower in Uganda compared to Scenario 1. In both scenarios, personnel time made up the majority of total program costs. In

**Table 5. PTBi program incremental costs in Kenya and Uganda (2020 $US).**

| | Scenario 1 (PTBi Study Implementation) | | Scenario 2 (Hypothetical MOH Model) | |
|---|---|---|---|---|
| | **Kenya** | **Uganda** | **Kenya** | **Uganda** |
| **Number of facilities*** | 9 | 4 | 17 | 6 |
| **PTBi package total** | $1,109,066 | $737,257 | $858,445 | $281,224 |
| **Per facility** | $123,229 | $184,314 | $50,497 | $46,871 |
| **Per birth (all phases)** | $48.13 | $17.19 | $23.91 | $5.47 |
| **Per birth (Phase 3)** | n/a | n/a | $16.36 | $3.47 |
| **Package cost by input type** | | | | |
| Personnel | | | | |
| Program-based | $416,669 | $274,898 | $158,214 | $46,758 |
| Specialist consultant | $280,183 | $241,319 | $107,631 | $93,593 |
| Facility-based | $254,985 | $72,236 | $490,717 | $65,570 |
| Office & simulation supplies | $32,694 | $31,441 | $30,622 | $26,489 |
| Capital goods | $13,070 | $12,214 | $21,146 | $13,170 |
| Prog. office/conf. space** | $13,008 | $10,776 | $21,319 | $5,467 |
| Travel | $99,231 | $96,053 | $28,779 | $30,176 |
| **Package cost by component** | | | | |
| General/non-allocable | $122,091 | $106,480 | $202,569 | $76,001 |
| DS | $56,183 | $45,034 | $88,283 | $26,753 |
| mSCC | $30,834 | $24,011 | $40,931 | $18,567 |
| PRONTO | $627,478 | $285,198 | $379,223 | $123,302 |
| QI | $272,479 | $256,534 | $147,421 | $36,599 |
| **Control cost per birth*** | | | | |
| Control (DS & mSCC only) | $15.22 | $6.13 | $6.78 | $1.59 |

*Scenario 1 (the PTBi study) included 9 intervention facilities in Kenya and 4 intervention facilities in Uganda; and 8 control facilities in Kenya and 2 control facilities in Uganda. In Scenario 2, we modeled costs for the intervention and control in all 17 facilities in Kenya and all 6 facilities in Uganda.

**Includes costs associated with program office/conference room space for implementation (non-research) activities for all phases, including phone service, the value of in-country office space used for non-research activities, office maintenance, and office utilities; also includes costs associated with renting hotel-based conference rooms for limited offsite meetings (1–2 per phase), including food and beverages served at meetings. Because the interventions do not require the construction, purchasing, or rental of new health/birth facility space, costs associated with health/birth facilities are not included in the analysis.

DS: data strengthening; mSCC: modified WHO safe childbirth checklist; PTBi: Preterm Birth Initiative; QI: Quality improvement.

Table 5 outlines total costs associated with the PTBi package of interventions for Scenarios 1 and 2. Costs are detailed by scenario, country, cost input category, and intervention component.

Scenario 2, 88% and 72% of costs were for personnel in Kenya and Uganda, respectively. Cost by input type, intervention component, and phase for Scenario 2 are in Fig 2 (Kenya) and Fig 3 (Uganda). Detailed costs by input type, phase, and package component are in S1 Table.

Among implementation phases in Scenario 2, Phase 1, 2, and 3 activities comprised 26%, 44%, and 30% of total costs in Kenya, respectively; in Uganda this was 38%, 37%, and 25%, respectively (S1 Table). Among intervention components, general/non-allocable costs were 24% of total costs in Kenya and 27% of total costs in Uganda; PRONTO costs were 44% of total costs in each Kenya and Uganda. Additional details on the relative contributions of each intervention component are in Table 6.

## Sensitivity analyses

Sensitivity analyses focused on areas of uncertainty in Scenario 2, identified through discussions with PTBi study team members and experts in the local healthcare systems in Kenya and

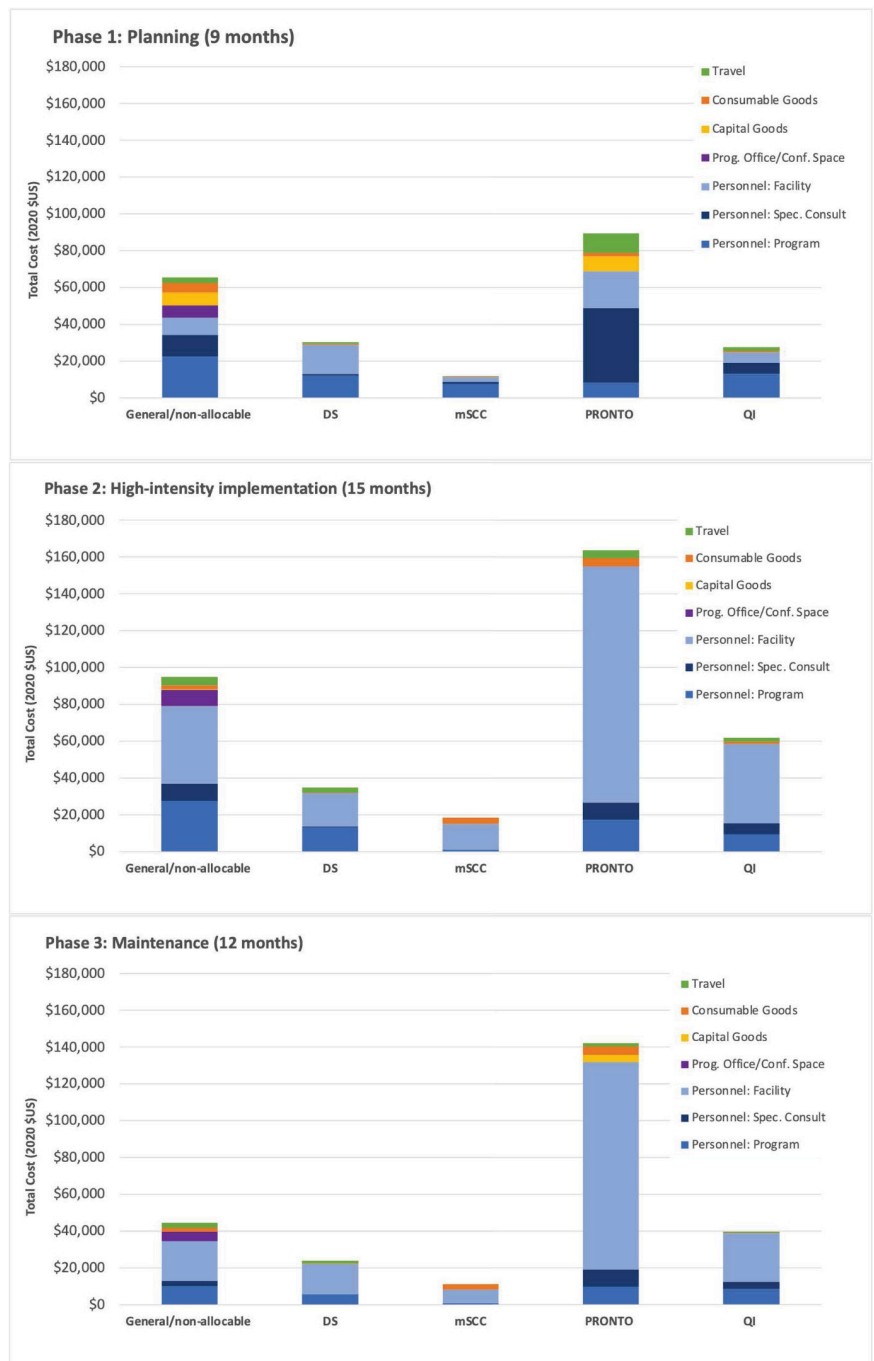

**Fig 2. Scenario 2 costs by input type and PTBi intervention component in Kenya (2020 $US).** DS: data strengthening; mSCC: modified WHO safe childbirth checklist; PTBi: Preterm Birth Initiative; QI: Quality improvement. Fig 2 depicts Scenario 2 (Hypothetical MOH Model) costs by phase and intervention component in Kenya for the PTBi package of interventions. Phases include: 1) program design and planning prior to in-facility implementation; 2) high-intensity initial implementation and training within facilities; and 3) annual maintenance, including refresher training and program monitoring.

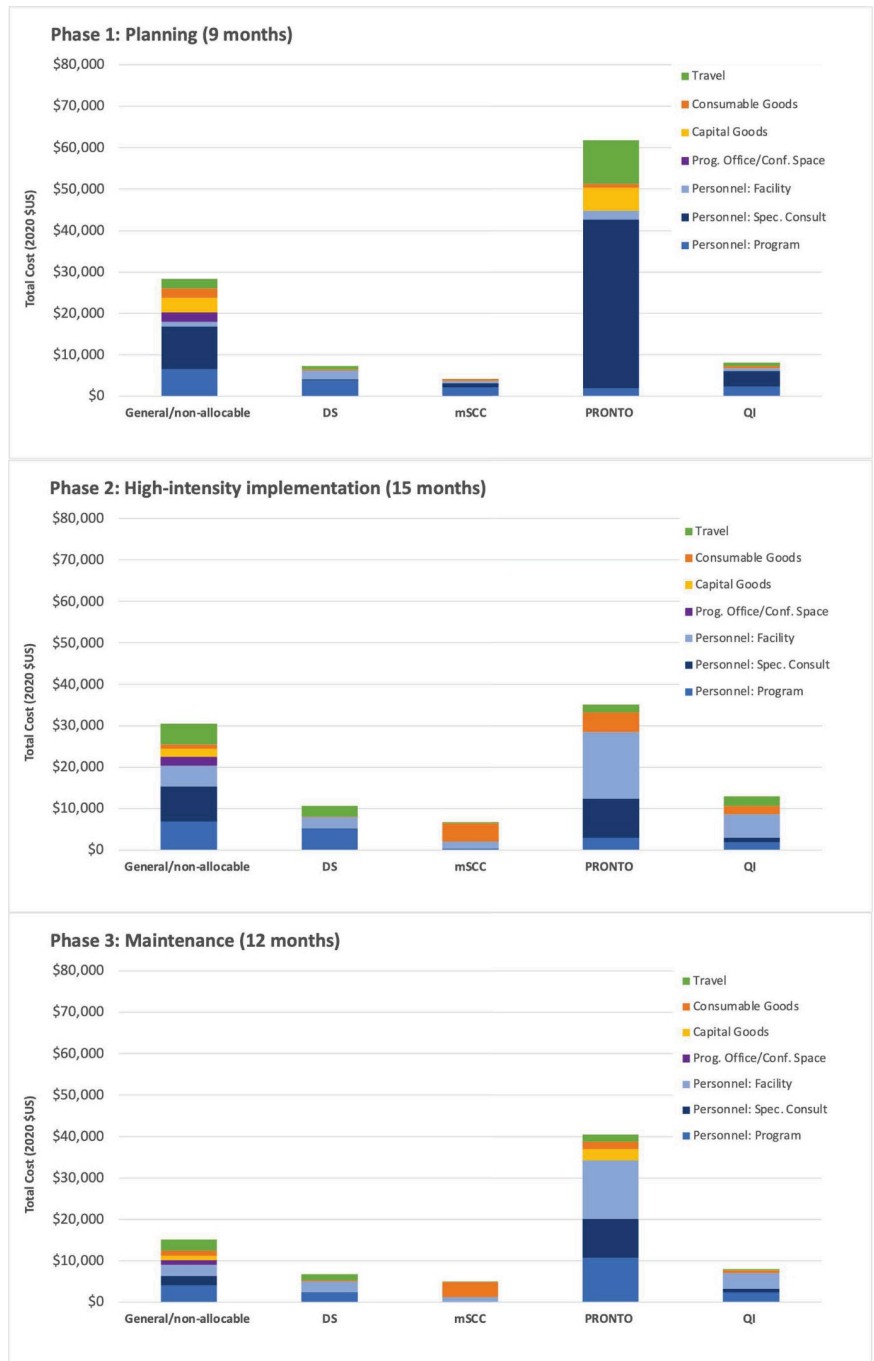

**Fig 3. Scenario 2 costs by input type and PTBi intervention component in Uganda (2020 $US).** DS: data strengthening; mSCC: modified WHO safe childbirth checklist; PTBi: Preterm Birth Initiative; QI: Quality improvement. Fig 3 depicts Scenario 2 (Hypothetical MOH Model) costs by phase and intervention component in Uganda for the PTBi package of interventions. Phases include: 1) program design and planning prior to in-facility implementation; 2) high-intensity initial implementation and training within facilities; and 3) annual maintenance, including refresher training and program monitoring.

**Table 6. Contributions of individual intervention components to total costs in Scenario 2.**

|  | Kenya | Uganda |
|---|---|---|
| **General/non-allocable** | 24% | 27% |
| **DS** | 10% | 10% |
| **mSCC** | 5% | 7% |
| **PRONTO** | 44% | 44% |
| **QI** | 17% | 13% |
| **Total** | **$858,445** | **$281,224** |

DS: data strengthening; mSCC: modified WHO safe childbirth checklist; PTBi: Preterm Birth Initiative; QI: quality improvement.

Table 6 outlines the percentage of the total program costs for each PTBi intervention component in Scenario 2. General/non-allocable costs included meetings and activities that were not specific to an individual intervention, such as stakeholder meetings to discuss the full package of interventions and pilot testing several components at once.

Note: some percentages may not add to 100% due to rounding.

Uganda. Key sensitivity analyses included differences in the intensity of program activities, including a higher level of program adaptation during Phase 1, more frequent in-facility trainings in Phase 2, and a different "model" for PRONTO Nurse Mentors (hiring NMs to travel to each facility, versus having NMs embedded within each facility). We also analyzed the effect of differences in health facility volumes between Kenya and Uganda to understand the impact of PTBi study facility type/volume between the two countries. Finally, we estimated total costs when excluding costs associated with facility-based workers.

Results from our sensitivity analyses are outlined in Table 7, reported in cost per birth and percent different from base case. Differences in PRONTO Nurse Mentor models resulted in a 4% increase ($24.92 vs $23.91) in cost per birth in Kenya and a 10% increase ($6.02 vs $5.47) in Uganda. A higher intensity of intervention adaptation activities resulted in increases in per-birth costs that were 8% higher in Kenya ($25.82 vs $23.91) and 22% higher in Uganda ($6.68 vs $5.47). Excluding in-facility health worker costs (except for in-facility Nurse Mentor training) resulted in costs that were 55% lower ($10.76 vs $23.91) in Kenya and 18% lower in Uganda ($4.48 vs $5.47) (see S2 Table for costs by input type, phase, and component). Implementation in fewer, higher-volume facilities in Kenya decreased total cost by 50% versus the base case ($11.90 vs $23.91); implementation in a greater number of lower-volume facilities in Uganda increased total costs by 49% vs the base case ($8.11 vs $5.47).

## Discussion

Using a combination of quantitative and qualitative methods, we estimated costs associated with the implementation of the PTBi package of interventions in two distinct scenarios: one that represents non-research costs observed in a study setting and one that represents hypothetical costs of an MOH-led model. We found that package costs were $48.13 per birth in Kenya in the PTBi Study and $17.19 per birth in Uganda (Scenario 1, all phases). In Scenario 2, costs per birth were $23.19 in Kenya and $5.47 in Uganda (all phases); steady-state costs for phase 3 alone were $16.36 per birth in Kenya and $3.47 in Uganda.

Our study adds to the limited evidence on the costs associated with interventions to improve preterm survival in lower-resource settings, where the burden of preterm birth is highest [1]. It is becoming increasingly accepted that complementary, multicomponent interventions—like the PTBi package—have more promise for preterm birth than interventions

**Table 7. Scenario 2 sensitivity analyses–variation in cost per birth in Kenya and Uganda, testing several assumptions.**

| Base case assumption | Alternative assumption | Kenya | | Uganda | |
|---|---|---|---|---|---|
| | | (base case: $23.91) | % diff. | (base case: $5.47) | % diff. |
| NMs hired from and embedded within facilities (one NM per facility)* | NMs travel to facilities | $24.92 | + 4% | $6.02 | + 10% |
| Minimal adjustments to package of interventions | Moderate adjustments (twice the effort of base case) | $25.82 | + 8% | $6.68 | + 22% |
| Maintenance phase:<br>• PRONTO activities 1 day/mo.<br>• QICs every 6 months | Maintenance phase:<br>• PRONTO activities 2 days/mo.<br>• QICs every 3 months | $27.49 | + 15% | $5.91 | + 8% |
| Six high-volume facilities in Uganda | Observed Uganda prices in mix of high- and low-volume facilities | n/a | — | $8.11 | + 49% |
| Mix of high- and low- volume facilities in Kenya (17) | Observed Kenya prices in only high-volume facilities | $11.90 | - 50% | n/a | — |
| Includes time spent by facility-based workers in intervention activities | Excludes health worker time | $10.76 | - 55% | $4.48 | - 18% |

*Our base assumption was that PRONTO activities within facilities would be led by a nurse, midwife, medical officer, or other qualified health professional from the facility who has been specially trained to lead PRONTO activities; when not leading PRONTO activities, the NMs would continue their regular roles within their facilities. This is in contrast to the alternative assumption, wherein nurse mentors would be hired to travel to facilities to lead PRONTO activities. In this alternative, one NM would be hired per 3–6 facilities, depending on facility volume, size, and distance.

NM: PRONTO Nurse Mentors; QICs: quality improvement collaboratives.

Table 7 provides an overview of sensitivity analyses for Scenario 2, including the base case assumption in the main analysis, alternative assumptions explored in sensitivity analyses, and results for each country. Results are presented in cost per birth and percent change from base case in cost per birth.

implemented individually [35]. Understanding the costs of intervention packages requires complex costing methods, including representation of shared costs for integrated programming and shared activities, understanding of potential synergies and efficiencies, and other contextual considerations, which we included in our study.

Due to limited data regarding the costs of interventions for preterm birth in lower-resource settings, differences in interventions, and differences in setting and results reporting, comparisons in costs between our analysis and other cost analysis in MNCH are limited. However, our input categories and unit costs are consistent with those in other cost analyses in similar settings for intrapartum care interventions [15, 36, 37], which may indicate that overall results are consistent with other studies and may be generalizable in Kenya and Uganda. Additionally, a 2015 analysis of facility-based health care costs found an average cost per birth of $106 USD in Kenya and $75 USD in Uganda [16]. After accounting for inflation, adding the PTBi package would increase the cost per delivery/birth by about 15% in Kenya and 5% in Uganda in Phase 3, unadjusted for offsetting and incurred costs of care for surviving preterm infants. This should be weighed against the high level of effectiveness of the program. Understanding the cost-effectiveness of the PTBi package of interventions—including downstream costs—and other similar multicomponent quality improvement packages, is an important area of future study.

Scenario 2 costs were substantially lower than Scenario 1 in both countries. This reflects differences in personnel pay, intensity of activities, and number of facilities in the two scenarios: Scenario 1 program-based staff were paid private-sector salaries/benefits; Scenario 1 includes time spent on full program and intervention development (higher intensity); and Scenario 2 includes more facilities (greater efficiency per facility and birth). The difference in Scenario 2 total costs and cost per birth are also due to differences in birth volumes in PTBi study facilities in Kenya and Uganda, which we used for estimating facility-level costs in both scenarios: In Kenya, most study facilities were smaller, lower-volume with a higher health-worker-to-birth

ratio; in Uganda, most study facilities were larger, with higher birth volumes and a lower health-worker-to-birth ratios. As we found in sensitivity analyses, implementation in larger, high-volume facilities (vs lower volume) results in greater efficiencies and lower cost per birth. This may be helpful for resource allocation, as more complicated deliveries often are referred to larger facilities when possible. However, that does not warrant the exclusion of smaller facilities from intervention activities. Maximizing impact and efficiency is relevant for resource allocation, but decision-making should also be based on burden and other needs.

In Scenario 2, total costs in Uganda were 33% of those in Kenya. This reflects the differences in the Kenyan and Ugandan economies and health care systems—Kenya has transitioned to being a lower-middle income country (GDP per capita in 2020: $1,838 USD), while Uganda remains a low-income country (GDP per capita in 2020: $817 USD) [38, 39]. As we observed through in-country data collection, public-sector health program officers and health workers in Kenya receive salaries that are two- to five-times higher than those in Uganda.

In our study, personnel were the single largest cost input, which reflects the fact that the PTBi package uses training and monitoring to improve health worker skills, including the identification and management of preterm birth. The package does not require the purchasing or maintenance of new high-tech goods or the implementation of new clinical interventions, which helps account for the relatively low estimated costs for consumable and capital goods. Our analyses included time spent by consultants who provide specialized guidance and training in clinical care and quality improvement. In Scenario 2, the total cost for specialist consultants was relatively comparable between Kenya and Uganda; this is due to anticipated similar pay for specialist consultants between the two countries.

Because we aimed for a robust economic evaluation, we included in-facility health worker time spent in training and QI activities in our analysis. While this allows us to understand the total value of the PTBi package of interventions and the opportunity costs of implementing health interventions, it is important to note that findings from our cost analysis is neither a budget estimate nor a financial analysis and should not be used as such. Removing costs associated with health professional time, given they are salaried employees already working in facilities, resulted in 20–50% lower total costs. Because our analysis includes time spent by existing health workers—the total "new" costs incurred by the PTBi package of interventions is potentially lower than estimated in our analysis.

PRONTO had the single largest cost among intervention components. This is in part due to the limited local experience with highly realistic simulation and team training offered by PRONTO, including training of trainers, which is led by US-based experts in simulation and team-based training. While the costs associated with PRONTO are substantial compared to that of the mSCC, for example, they represent the intensive, skills-based, capacity-building nature of the PRONTO intervention. PRONTO has been demonstrated in several studies to significantly improve maternal and neonatal care [31, 40, 41] and has been met with much enthusiasm by health workers and local stakeholders in the PTBi study and other evaluations [21, 42]. However, success of the PRONTO intervention relies on continued use of simulations and team training activities—which we modeled in our analysis—and long-term investments in adding new curricular components and training of national-level cohorts of master trainers (not observed in the PTBi study or included in our analysis). Larger scale implementation of the PRONTO intervention is underway in India and other countries in sub-Saharan Africa; further analysis of PRONTO at scale and using different nurse mentor models is an important area of further study.

Our analysis has important limitations. First, since Scenario 2 is hypothetical, we did not directly observe costs for this scenario, and we relied on several assumptions, including heavy use of public sector personnel and minimal adaptation of package interventions. Actual in-

country costs may differ, as outlined in our sensitivity analyses, and the effectiveness of the PTBi package of interventions and active control in Scenario 2 may differ from the PTBi study. We relied on publicly available information from in-country governments regarding public-sector health worker pay, especially in Scenario 2; any differences between publicly available salary information and actual health worker pay may affect our estimates. However, our assumptions were based on information from in-country researchers, health workers, and healthcare system experts. Changes in key model assumptions resulted increases in cost per birth of 3–22% in sensitivity analyses.

Second, our costs represent implementation in a county-specific context in Kenya and a region-specific context in Uganda over the course of 3 years. Our two scenarios and their associated costs are also based on information from the specific facilities included in the PTBi study. Scaling at a national level may result in modest efficiencies, though costs and resources required to plan, implement, and maintain the PTBi package may differ in other facilities. Third, the PRONTO implementation model in our study is focused on preterm birth and select obstetric emergencies. Broader PRONTO models covering more clinical situations would cost more.

Strengths of our study include its high level of detail, robust data collection, and in-depth analysis of costs by activity, input type, phase, and intervention component. Our modeling of scenarios provides insights into both private-sector and public-sector costs, and our analysis approach can be applied to other settings. While costs in our study are specific to local contexts in Kenya and Uganda, our methods may be of use for understanding costs in other geographic settings. Additionally, the analysis approach and our use of national data for health worker composition and costs may be applied to other clinical situations related to obstetric and neonatal care, using the same intervention components (DS, mSCC, PRONTO, and QI). Indeed, we found that costs for PTBi intervention maintenance were lower than for previous phases, which indicates that PTBi may be a suitable platform for the efficient inclusion additional training and curricula in other areas of emergency obstetric, intrapartum, and neonatal care. This is another important area that should be explored in future economic evaluations.

## Conclusions

We found that the PTBi package may be implemented and maintained by the public sectors in Kenya and Uganda at a relatively low cost per birth. While our study is limited by its assumptions and specific setting and context, our findings add to the limited data regarding the total economic costs associated with interventions to improve preterm survival in lower-resource settings. Such findings may help inform future policy and implementation of this critical, life-saving package of interventions.

## Supporting information

**S1 Fig. Detailed PTBi phases and activities for Scenario 2 cost analysis.** S1 Fig provides detailed timing, phasing, and activities for each of the PTBi package intervention components through all phases of program planning, initial high-intensity implementation, and maintenance. All activities are based on activities for and findings from the PTBi study. All activities and durations were used for the Scenario 2 cost analysis. Control costs included only activities for data strengthening and the WHO modified safe childbirth checklist, along with general program activities that pertained to these two components. Intervention package costs included all activities for data strengthening, mSCC, PRONTO, Quality Improvement, and general program activities.
(XLSX)

**S2 Fig. Detailed PTBi costing assumptions for Scenario 2.** Detailed model assumptions for PTBi costing Scenario 2.
(XLSX)

**S1 Table. Scenario 2 total PTBi package costs by phase and intervention component.**
(XLSX)

**S2 Table. Scenario 2 PTBi package costs excluding facility-based personnel.**
(XLSX)

**S1 Checklist. Inclusivity in global research.**
(DOCX)

## Acknowledgments

We thank the Bill & Melinda Gates Foundation for their generous funding, and our program officers for their guidance, encouragement, and support. We would like to thank those who were critical to the PTBi study, including the mothers and infants who participated in the study; the healthcare providers and other staff working in study facilities; and the Ministry of Health officials and advisory boards who supported and guided implementation and research efforts. Furthermore, we thank the research teams and implementors at Makerere University, Kenya Medical Research Institute, UCSF, and PRONTO International who provided invaluable insights and input into this cost analysis specifically. We are grateful for the members of the Preterm Birth Initiative East Africa External Advisory Committee, our colleagues at the California Preterm Birth Initiative, and the members of our joint Strategic Advisory Board.

## Author Contributions

**Conceptualization:** Carolyn Smith Hughes, Elizabeth Butrick, Juliana Namutundu, Phelgona Otieno, Peter Waiswa, Dilys Walker, James G. Kahn.

**Data curation:** Carolyn Smith Hughes, Elizabeth Butrick, Juliana Namutundu, Easter Olwanda, Peter Waiswa, Dilys Walker, James G. Kahn.

**Formal analysis:** Carolyn Smith Hughes, Elizabeth Butrick, Juliana Namutundu, Easter Olwanda, Dilys Walker, James G. Kahn.

**Funding acquisition:** Elizabeth Butrick, Phelgona Otieno, Peter Waiswa, Dilys Walker.

**Investigation:** Elizabeth Butrick, Phelgona Otieno, Peter Waiswa, Dilys Walker.

**Methodology:** Carolyn Smith Hughes, Elizabeth Butrick, Juliana Namutundu, Easter Olwanda, Phelgona Otieno, Peter Waiswa, Dilys Walker, James G. Kahn.

**Project administration:** Elizabeth Butrick, Dilys Walker.

**Resources:** Elizabeth Butrick.

**Software:** Carolyn Smith Hughes.

**Supervision:** Carolyn Smith Hughes, Elizabeth Butrick, Phelgona Otieno, Peter Waiswa, Dilys Walker, James G. Kahn.

**Validation:** Carolyn Smith Hughes, Elizabeth Butrick, Juliana Namutundu, Easter Olwanda, Phelgona Otieno, Peter Waiswa, Dilys Walker, James G. Kahn.

**Visualization:** Carolyn Smith Hughes, Elizabeth Butrick, Juliana Namutundu, Easter Olwanda, Dilys Walker, James G. Kahn.

**Writing – original draft:** Carolyn Smith Hughes, Elizabeth Butrick, Dilys Walker, James G. Kahn.

**Writing – review & editing:** Carolyn Smith Hughes, Elizabeth Butrick, Juliana Namutundu, Easter Olwanda, Phelgona Otieno, Peter Waiswa, Dilys Walker, James G. Kahn.

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
