## [Decision Letter · Decision Letter 0]

2 Jun 2022

PONE-D-22-11311Estimated costs for an intrapartum quality improvement package for improving preterm survival and reinforcing best practices in Kenya and UgandaPLOS ONE

Dear Dr. Smith Hughes,

Thank you for submitting your manuscript to PLOS ONE. After careful consideration, we feel that it has merit but does not fully meet PLOS ONE’s publication criteria as it currently stands. Therefore, we invite you to submit a revised version of the manuscript that addresses the points raised during the review process.

We look forward to receiving your revised manuscript.

Kind regards,

Dylan A Mordaunt, MD, MPH, FRACP

Academic Editor

PLOS ONE

Journal Requirements:

"DW is cofounder of PRONTO International. The other authors declare no competing interests."

Additional Editor Comments:

Thank you for your submission. We received relatively positive reviews with what are essentially minor revisions though characterised as major revisions.

With regards to the criteria for publication:

1. The study appears to present the results of original research.

2. Results reported appear not to have been published elsewhere.

3. Experiments, statistics, and other analyses are performed to a high technical standard. Additional detail and clarificaiton should be made as per the reviewers.

4. Conclusions are presented in an appropriate fashion and are supported by the data.

5. The article is presented in an intelligible fashion and is written in standard English.

6. The research meets all applicable standards for the ethics of experimentation and research integrity.

7. The article adheres to appropriate reporting guidelines and community standards for data availability. There is currently no checklist for costing studies, however the CHEERS checklist contains relevant information. It would be worth clarifying whether all relevant components have been included.

I look forward to receiving your resubmission.

Reviewers' comments:

Reviewer's Responses to Questions

**Comments to the Author**

1. Is the manuscript technically sound, and do the data support the conclusions?

Reviewer #1: Yes

Reviewer #2: Yes

Reviewer #3: Yes

2. Has the statistical analysis been performed appropriately and rigorously? 

Reviewer #1: Yes

Reviewer #2: Yes

Reviewer #3: Yes

3. Have the authors made all data underlying the findings in their manuscript fully available?

Reviewer #1: Yes

Reviewer #2: Yes

Reviewer #3: Yes

4. Is the manuscript presented in an intelligible fashion and written in standard English?

Reviewer #1: Yes

Reviewer #2: Yes

Reviewer #3: Yes

5. Review Comments to the Author

Reviewer #1: Thank you for the opportunity to review this manuscript. I liked the paper a lot and feel that the methodology and approach will be really useful for other researchers to use as it sets out a very clear and logical approach.

There are some points of clarification needed – especially in the Abstract which I felt did not well reflect the paper. Recognizing that the Abstract is all that many people read, I feel it could be clearer.

The abstract talks about an intervention and an active control. I think it would be more useful to say that one was the non-research costs associated with a study and one was implementation by the Ministry of Health. That feels clearer than an active control which suggests a trial.

The Preterm Birth Initiative East Africa needs to be mentioned in the Introduction to the Abstract. At the moment the intro is about the Safe Childbirth check and then the next sentence is about the PTBi. There is no clear connection and this could be stronger and better linked.

It is really helpful to have the costs for each scenario but what was missing for me was the effectiveness. Was it the same in each? Did the lesser cost result in less effectiveness? I know this is probably another study but it feels like a significant gap.

The authors say that the PTBi costs are relatively low cost to implement but relative to what? For many countries, a cost of $23 per birth for an education and support program would be more than is spent on the woman’s care overall? I would like some more about how these costs compare relative to the overall costs.

Reviewer #2: For the sensitivity analysis using scenario2, please call it hypothetical model and avoid using the term implementation. Overall the paper is good and may add to the body of knowledge on costing studies of PTBi interventions.

Reviewer #3: This manuscript gives a detailed account about the costs of a complex intervention package aiming to improve preterm bird outcomes (PTBi) in Kenya and Uganda. From actual costs of a cluster-randomised trial, costs of real-world implementation were projected. A full and a minimum implementation scenario was compared.

Overall, the manuscript is well written, reporting is detailed, methods and assumptions are well justified. Nevertheless, some important questions need clarification and the below suggestions may help the authors to improve the manuscript.

a) Please clarify the decision situation and implications on real-life care: it was not clear, if the program implementation changes the costs of actual practice (e.g. if more resources are used after implementing the mSCC tool) and whether it is taken into account in cost calculations.

b) I suggest the authors briefly introduce the essential content and clinical evidence concerning the interventions of the package. External references were provided, 1-2 more sentences would help the reader to understand the program

c) Please briefly justify that the PTBi study hospitals reflected the national average and extensions to national costs can be made.

e) I suggest that in addition to main unit costs, the authors provide details of resource use by each study component, and link assumptions / scenarios to actual resource use figures. The paper provides lengthy explanations about various assumptions, but it remains unclear, how they were operationalised in the analysis.

f) It was not entirely clear, why minimal implementation was chosen as control. In my opinion, incremental costs vs current practice would be more relevant. The authors should briefly describe current practice in the index countries, and if any elements of the PTBi package are being spontaneously implemented across the countries

g) In the results tables and figures it was difficult to follow if total costs or incremental costs vs control were provided or where the incremental costs vs control are detailed.

h) When calculating costs per birth, did the authors project costs to live births or per delivery (resulting in live births and still births)? Were the expected effects of the program on live births taken into consideration when projecting costs per birth? While the paper is not a full economic analysis, the authors may consider projecting costs per natural outcome such as cost per prevented stillbirth or cost per baby surviving one year etc...

i) the authors should comment about the uncertainty of their estimates

6. PLOS authors have the option to publish the peer review history of their article (what does this mean?). If published, this will include your full peer review and any attached files.

Reviewer #1: No

Reviewer #2: **Yes: **Roopali Goyanka

Reviewer #3: **Yes: **Zsombor Zrubka, MD, MBA, PhD

---

## [Author Response · Author response to Decision Letter 0]

22 Dec 2022

Journal Requirements:

• Thank you for providing these links. We have reviewed the style requirements and revised our track-changes file (and therefore also in the clean manuscript file as well), to make sure our files comply with them.

• Thank you. We have completed the form and attached it in the submission portal along with our other files.

"DW is cofounder of PRONTO International. The other authors declare no competing interests."

• We have confirmed that DW’s role with PRONTO International does not alter our adherence to PLOS ONE policies on data sharing. 

• We have included the updated text in the Cover Letter attached to our submission.

• Thank you for this information. We have begun the process of preparing our data for submission to the UCSF Dryad repository and will provide it should the manuscript be accepted for submission. 

o Our dataset "PTBi Cost Analysis Data" has been uploaded

o The DOI and URL are as follow: 

o DOI https://doi.org/10.7272/Q64X562W

o We have provided this information in the cover letter as requested.

 

Additional Editor Comments:

Thank you for your submission. We received relatively positive reviews with what are essentially minor revisions though characterised as major revisions.

• Thank you for the thorough review, comments, and opportunity to revise and resubmit our manuscript. 

With regards to the criteria for publication:

1. The study appears to present the results of original research.

• Confirmed: Our study/research is original and has not been conducted by other researchers at the time of submission. 

2. Results reported appear not to have been published elsewhere.

• Confirmed: Our results have not been reported or published in other publications. 

3. Experiments, statistics, and other analyses are performed to a high technical standard. Additional detail and clarificaiton should be made as per the reviewers.

• We’ve provided additional details, clarifications, and revisions to the manuscript based on comments from the reviewers, as described below. 

4. Conclusions are presented in an appropriate fashion and are supported by the data.

• Thank you for reviewing; based on this comment we did not make specific edits.

5. The article is presented in an intelligible fashion and is written in standard English.

• Thank you for reviewing; based on this comment we did not make specific edits.

6. The research meets all applicable standards for the ethics of experimentation and research integrity.

• The PTBi study was reviewed and approved by US-based and local institutional review boards. We have provided details regarding reviews and approvals and other ethical considerations in the first paragraph of the “Study setting” section of our manuscript. Based on this comment we did not make specific edits.

7. The article adheres to appropriate reporting guidelines and community standards for data availability. There is currently no checklist for costing studies, however the CHEERS checklist contains relevant information. It would be worth clarifying whether all relevant components have been included.

• We have reviewed the applicable portions of the CHEERS checklist (ie, those that apply to cost analyses). We confirm that we have revised to manuscript to ensure that all relevant components from the CHEERS checklist have been included in our manuscript. 

I look forward to receiving your resubmission.

 

Reviewers' comments:

Reviewer's Responses to Questions

Comments to the Author

1. Is the manuscript technically sound, and do the data support the conclusions?

Reviewer #1: Yes

Reviewer #2: Yes

Reviewer #3: Yes

• Thank you for reviewing; based on this we did not make specific edits.

2. Has the statistical analysis been performed appropriately and rigorously?

Reviewer #1: Yes

Reviewer #2: Yes

Reviewer #3: Yes

• Thank you for reviewing; based on this we did not make specific edits.

3. Have the authors made all data underlying the findings in their manuscript fully available?

Reviewer #1: Yes

Reviewer #2: Yes

Reviewer #3: Yes

• Thank you for reviewing; based on this we did not make specific edits.

4. Is the manuscript presented in an intelligible fashion and written in standard English?

Reviewer #1: Yes

Reviewer #2: Yes

Reviewer #3: Yes

• Thank you for reviewing; based on this we did not make specific edits.

5. Review Comments to the Author

 

Reviewer #1: Thank you for the opportunity to review this manuscript. I liked the paper a lot and feel that the methodology and approach will be really useful for other researchers to use as it sets out a very clear and logical approach.

• Thank you for your thorough review and extremely helpful comments. We made revisions as appropriate to the manuscript, as detailed below. 

There are some points of clarification needed – especially in the Abstract which I felt did not well reflect the paper. Recognizing that the Abstract is all that many people read, I feel it could be clearer.

• We have revised the abstract based on your helpful comments and requests for clarification, and to ensure it better reflects the paper. Further details on specific changes are listed in the specific comments below.

The abstract talks about an intervention and an active control. I think it would be more useful to say that one was the non-research costs associated with a study and one was implementation by the Ministry of Health. That feels clearer than an active control which suggests a trial.

• Thank you for the opportunity to clarify and your comments. In the PTBi effectiveness study, the PTBi package of interventions was compared to the more minimal active control. In this cost analysis, we calculated costs associated with the PTBi package of interventions and the active control in two scenarios: 

o Our first scenario reflects the observed non-research costs within the PTBi study (for the PTBi package of interventions and active control), which includes many private-sector costs that would not be incurred if the program was implemented in the public sector.

o Our second scenario estimates the hypothetical costs for non-study public-sector implementation (Ministry of Health implementation) of the PTBi package of interventions compared to the active control.

o We have worked to make this clearer in the abstract and throughout the full manuscript as well. 

• We have provided more detail about the process used in the PTBi study to develop the package of interventions and active control, and our rationale for the inclusion of the active control in the cost analysis (full submission PDF pages 68-69, lines 163-192; page 70, lines 208-212; and pages 71-72, lines 235-243). We have also worked to make this more clear in the abstract. 

The Preterm Birth Initiative East Africa needs to be mentioned in the Introduction to the Abstract. At the moment the intro is about the Safe Childbirth check and then the next sentence is about the PTBi. There is no clear connection and this could be stronger and better linked.

• Thank you for this comment; we have revised the abstract accordingly to more clearly describe the PTBi intervention and active control (and components within them) in the introduction to the abstract.

It is really helpful to have the costs for each scenario but what was missing for me was the effectiveness. Was it the same in each? 

• The effectiveness of the intervention and control in Scenario 1 are known and are mentioned in the second paragraph of the “Study Setting” section (34% lower odds of stillbirth and early neonatal death for the PTBi package vs the active control). 

• The effectiveness of the PTBi package of interventions versus active control in Scenario 2 – hypothetical MOH model – was not observed and is not known. We have clarified this caveat in the limitations in the discussion on PDF page 92, lines 702-712. However, the frequency of activities and types of activities associated with each of the intervention components is assumed be similar, and, as we note in the manuscript, our assumptions for Scenario 2 were developed in collaboration with in-country experts in program implementation and costs.

• We have revised the full text manuscript to include some additional details regarding the process by which the PTBi package of interventions was developed, and its effectiveness and feasibility on PDF page 67-68, lines 151-158.

Did the lesser cost result in less effectiveness? I know this is probably another study but it feels like a significant gap.

• Because the focus of this analysis is on costs, the effectiveness details have been published elsewhere and are referenced in full text, and given the limited space in the abstract, we have not included a detailed description of the effectiveness in the abstract or main text. 

• As you’ve noted, the effectiveness of the PTBi package is an extremely important consideration. We plan to assess the cost-effectiveness of the PTBi package vs active control (for both Scenarios) – including under potentially lower effectiveness in Scenario 2 due to differences in implementation model – in a forthcoming cost-effectiveness analysis that will integrate the health effects/health outcomes of the PTBi package of interventions vs active control in both scenarios. 

The authors say that the PTBi costs are relatively low cost to implement but relative to what? For many countries, a cost of $23 per birth for an education and support program would be more than is spent on the woman’s care overall? I would like some more about how these costs compare relative to the overall costs.

• We really appreciate this point, and we have removed the mention of “low” in the abstract and discussion. We had intended to mean “low” compared to the cost of delivery, but we appreciate that such a statement is subjective and requires more information. 

• In addition to removing the word “low” from the abstract and discussion: We have revised the Introduction in the manuscript to include the estimated costs of maternity care in sub-Saharan African countries to provide some context and information by which to compare the costs of the PTBi intervention to overall costs of current practice (PDF page 62, lines 108-115). We also provide details on the current cost per delivery for maternity care in each country in the discussion on PDF page 90, lines 635-637. We hope this sufficiently answers the question regarding how the costs compare to overall costs of maternity care in the region.

 

Reviewer #2: For the sensitivity analysis using scenario2, please call it hypothetical model and avoid using the term implementation. Overall the paper is good and may add to the body of knowledge on costing studies of PTBi interventions.

• Thank you for this opportunity to clarify. We have revised the wording/labelling accordingly throughout the paper, using “hypothetical MOH model” in most instances where we refer to Scenario 2, especially in tables and visuals (Tables 2-5). 

 

Reviewer #3: This manuscript gives a detailed account about the costs of a complex intervention package aiming to improve preterm bird outcomes (PTBi) in Kenya and Uganda. From actual costs of a cluster-randomised trial, costs of real-world implementation were projected. A full and a minimum implementation scenario was compared.

Overall, the manuscript is well written, reporting is detailed, methods and assumptions are well justified. Nevertheless, some important questions need clarification and the below suggestions may help the authors to improve the manuscript.

• Thank you for your comments and questions. We feel they helped us greatly in improving the study and manuscript. Our responses to each of your comments are below. 

a) Please clarify the decision situation and implications on real-life care: it was not clear, if the program implementation changes the costs of actual practice (e.g. if more resources are used after implementing the mSCC tool) and whether it is taken into account in cost calculations.

• Thank you for this question. Our analysis focuses on the costs of implementing the activities outlined within the PTBi package of interventions, which includes health worker training, data strengthening, and other activities that reinforce the use of evidence-based practices and care per guidelines. However, because the PTBi intervention does not include the use of new medical interventions that are not part of the current standard of care, we did not include costs associated with most changes to direct patient care in our study. For example, we did include the additional time it would take for health workers to fill out the mSCC while providing care, but we did not include costs associated with downstream changes in care. 

• To help clarify, we have provided additional details on the focus of our cost analysis in the Abstract; the details of the PTBi package of interventions and control on PDF page 67, lines 132-140; PDF page 68, lines 161-176; PDF pages 71-72, lines 235-243; and page 78, lines 356-362. 

• We will be taking into account the changes in the costs of actual practices in a forthcoming cost-effectiveness analysis. While we do assume that implementation of the PTBi intervention and active control would have an impact on the costs of actual practices, the affect may increase some costs and may decrease others. For example, the package promotes the use of antenatal corticosteroids (ACS, per guidelines) for lung maturation in preterm infants, so implementation of the package would result in greater use—and associated costs—of ACS. However, when used appropriately, ACS can also avert costs associated with the need for ventilation and use of more intensive care among some infants. Our planned cost-effectiveness analysis takes these changes in clinical care and associated costs and health outcomes, including longer-term costs and outcomes, into account. 

b) I suggest the authors briefly introduce the essential content and clinical evidence concerning the interventions of the package. External references were provided, 1-2 more sentences would help the reader to understand the program

• We have revised the manuscript to include further details regarding the package of interventions – including information on clinical evidence for each intervention component and their inclusion in the package of interventions and control – in the track-changes portion of the manuscript (PDF pages 67-69, lines 151-192), including the process behind and rationale for their selection. 

c) Please briefly justify that the PTBi study hospitals reflected the national average and extensions to national costs can be made.

• In a peer-reviewed publication regarding the design and implementation of the PTBi study, the selection of study facilities is described. The paper notes that “Given that facilities were not selected from a target population of hospitals, the intervention effects should be interpreted as impact evaluation of the intervention package implemented at the said facilities.” This is also true of results from our cost analysis, as we note in the revised limitations of our track-changes manuscript on PDF page 93, lines 714-718. 

• However, the PTBi study included facilities that were operated through the local public sector MNCH programming offices, which are under the purview of the national MOH. We reviewed organizational charts, staffing guidelines, and reporting structures to ensure our estimates for the composition of health workers in the study facilities – health worker costs made up the largest proportion of costs in our study – were representative of those outlined in national recommendations. 

• Further, we would like to note that, in Scenario 2, we used national data for all in-country programmatic and facility-based health worker salaries, and, where possible, national data from public-sector tenders, WHO pricelists, and UNICEF pricelists for resources used. To the best of our knowledge, these costs represent national costs. Study co-authors collaboratively collected and validated data to ensure their representativeness and to ensure they were appropriate for use in our study.

e) I suggest that in addition to main unit costs, the authors provide details of resource use by each study component, and link assumptions / scenarios to actual resource use figures. The paper provides lengthy explanations about various assumptions, but it remains unclear, how they were operationalised in the analysis.

• Thank you for this comment. We agree that the units for resource use are important. We have added units for major inputs into Table 3 for key resources. Full details on units per input and by each component are in our full data set, available through the linked data repository and available upon reasonable request. 

f) It was not entirely clear, why minimal implementation was chosen as control. In my opinion, incremental costs vs current practice would be more relevant. The authors should briefly describe current practice in the index countries, and if any elements of the PTBi package are being spontaneously implemented across the countries

• Thank you for this question/note. Regarding the question about why minimal implementation was the control: We costed the active control as a comparator because it was the comparator in the effectiveness portion of the PTBi study. For reference, this active control will also be used to assess cost-effectiveness, which is also a reason why we costed the control in our analysis. 

• To make this clearer in the manuscript – this is an important clarification that you’ve raised – we’ve added the rationale for the use of the active control in the effectiveness study and the cost analysis to the paper on PDF page 69, lines 187-192; and page 70, lines 208-212. 

• Regarding the comment about the current practice: In conjunction with changes outlined above, we have noted that prior to the PTBi study, none of the PTBi package or control components were being consistently implemented (PDF page 68, lines 173-176). 

• Full details on the current clinical practices in obstetric care have been added to the manuscript as well, which helps provide rationale for the PTBi package and the active control (pages 68-69, lines 163-185). We included some contextual information regarding costs of current practice from other studies on PDF page 66, lines 108-115.

g) In the results tables and figures it was difficult to follow if total costs or incremental costs vs control were provided or where the incremental costs vs control are detailed.

• All costs reported (for each the intervention package and active control) are incremental to the current practice; in our study, the current practice was not costed, but was assumed to be the current costs of providing intrapartum care for mothers during labor, delivery, and the immediate postpartum period and care for newborns during the delivery admission. 

• We have clarified in the Abstract, Introduction, and Methods that the goal of our study is to calculate incremental costs; these changes are specifically detailed in the track-changes file on PDF page 63, lines 52-53; page 67, lines 132-137; pages 71-72, lines 235-243; page 81-82, lines 450-459. 

h) When calculating costs per birth, did the authors project costs to live births or per delivery (resulting in live births and still births)? 

• We calculated the costs per delivery/birth (inclusive of both live births and stillbirths), regardless of neonatal outcome. We have made this clearer in the manuscript, for example on PDF pages 67, lines 130-131. we have also replaced the mentions of “infants” and “babies” where relevant in the manuscript and used “births” or “deliveries”. 

Were the expected effects of the program on live births taken into consideration when projecting costs per birth? While the paper is not a full economic analysis, the authors may consider projecting costs per natural outcome such as cost per prevented stillbirth or cost per baby surviving one year etc...

• We did not take into account the expected effects of the program on live births vs stillbirth, since we will be evaluating cost per health outcome (cost per stillbirth averted and cost per DALY averted for intervention package versus active control) in a forthcoming cost-effectiveness analysis that will be a follow-up to this study.

i) the authors should comment about the uncertainty of their estimates

• We have updated our discussion section to further describe our uncertainties in the limitations section of our discussion on PDF pages 92-93 in lines 702-720 – including more details regarding our uncertainties in Scenario 2. Our sensitivity analyses explore the effects of uncertainty in many inputs, as described on PDF pages 87-88 in the manuscript and Table 7.

---

## [Decision Letter · Decision Letter 1]

4 Jun 2023

Cost analysis of an intrapartum quality improvement package for improving preterm survival and reinforcing best practices in Kenya and Uganda

PONE-D-22-11311R1

Dear Dr. Carolyn Smith Hughes

We’re pleased to inform you that your manuscript has been judged scientifically suitable for publication and will be formally accepted for publication once it meets all outstanding technical requirements.

Kind regards,

Fadhlun Alwy Al-beity, MMed, PhD (ongoing)

Academic Editor

PLOS ONE

Additional Editor Comments (optional):

Please check minor format issues eg. line 158 in the clean manuscript

Reviewers' comments:

Reviewer's Responses to Questions

**Comments to the Author**

1. If the authors have adequately addressed your comments raised in a previous round of review and you feel that this manuscript is now acceptable for publication, you may indicate that here to bypass the “Comments to the Author” section, enter your conflict of interest statement in the “Confidential to Editor” section, and submit your "Accept" recommendation.

Reviewer #1: All comments have been addressed

2. Is the manuscript technically sound, and do the data support the conclusions?

Reviewer #1: Yes

3. Has the statistical analysis been performed appropriately and rigorously? 

Reviewer #1: Yes

4. Have the authors made all data underlying the findings in their manuscript fully available?

Reviewer #1: Yes

5. Is the manuscript presented in an intelligible fashion and written in standard English?

Reviewer #1: Yes

6. Review Comments to the Author

Reviewer #1: My comments have been addressed. I have no further changes to request. This paper will be useful to others especially in similar countries

7. PLOS authors have the option to publish the peer review history of their article (what does this mean?). If published, this will include your full peer review and any attached files.

Reviewer #1: **Yes: **Caroline Homer

---

## [Editor Report · Acceptance letter]

13 Jun 2023

PONE-D-22-11311R1 

Cost analysis of an intrapartum quality improvement package for improving preterm survival and reinforcing best practices in Kenya and Uganda 

Dear Dr. Smith Hughes:

I'm pleased to inform you that your manuscript has been deemed suitable for publication in PLOS ONE. Congratulations! Your manuscript is now with our production department. 

Kind regards, 

on behalf of

Dr. Fadhlun Alwy Al-beity 

Academic Editor

PLOS ONE